# Prognostic Gene Discovery in Glioblastoma Patients using Deep Learning

**DOI:** 10.3390/cancers11010053

**Published:** 2019-01-08

**Authors:** Kelvin K. Wong, Robert Rostomily, Stephen T. C. Wong

**Affiliations:** 1Department of Systems Medicine and Bioengineering, Houston Methodist, Houston, TX 77030, USA; stwong@houstonmethodist.org; 2Department of Neurological Surgery, Weill Cornell Medicine, New York, NY 10065, USA; 3Department of Radiology, Weill Cornell Medicine, New York, NY 10065, USA; 4Department of Neurosurgery, Houston Methodist Neurological Institute, Houston, TX 77030, USA; rrostomily@houstonmethodist.org; 5Department of Neuroscience, Weill Cornell Medicine, New York, NY 10065, USA; 6Department of Pathology and Laboratory Medicine, Weill Cornell Medicine, New York, NY 10065, USA

**Keywords:** deep learning, discovery, glioblastoma, glioblastoma stem cells, survival prediction

## Abstract

This study aims to discover genes with prognostic potential for glioblastoma (GBM) patients’ survival in a patient group that has gone through standard of care treatments including surgeries and chemotherapies, using tumor gene expression at initial diagnosis before treatment. The Cancer Genome Atlas (TCGA) GBM gene expression data are used as inputs to build a deep multilayer perceptron network to predict patient survival risk using partial likelihood as loss function. Genes that are important to the model are identified by the input permutation method. Univariate and multivariate Cox survival models are used to assess the predictive value of deep learned features in addition to clinical, mutation, and methylation factors. The prediction performance of the deep learning method was compared to other machine learning methods including the ridge, adaptive Lasso, and elastic net Cox regression models. Twenty-seven deep-learned features are extracted through deep learning to predict overall survival. The top 10 ranked genes with the highest impact on these features are related to glioblastoma stem cells, stem cell niche environment, and treatment resistance mechanisms, including *POSTN*, *TNR*, *BCAN*, *GAD1*, *TMSB15B*, *SCG3*, *PLA2G2A*, *NNMT*, *CHI3L1* and *ELAVL4*.

## 1. Introduction

Deep learning [1,2,3,4,5,6,7] has been used to learn prognostic subtypes of glioblastoma using pan-cancer gene expression data from The Cancer Genome Atlas (TCGA) [8], predict drug synergy based on cancer cell gene expression data [9], and predict survival based on multi-omics integrated data in liver cancer [10] etc. [11,12,13]. It is important to explain the model in a meaningful way to understand the deep learning model and its limitations. A typical deep learning model involves millions of parameters, which makes it a difficult task to understand. We propose to use feature importance ranking within the deep learning model. While feature importance ranking is popular with machine learning [14], its use within the deep learning model is rare, especially in cancer genomics where a model usually includes thousands of features. In this paper, we expand the permutation feature importance techniques to deep learning. Our goal is to study the inside of a trained deep learning model to discover prognostic gene features in glioblastoma.

Conventional machine learning approaches have been used to determine the gene expression that are prognostic to glioblastoma patient survival [15,16,17]. Glioblastoma gene expression regression modeling using the least absolute shrinkage and selection operator (Lasso) flavor strategy performed better when cancer pathway genes were used as input variables compared to whole genome input [16]. However, these types of penalized regression methods often require dropping a large number of genes in order to fit the survival outcome, hindering biological pathway interpretation and introducing random bias toward the selected gene factors. Deep learning offers the capacity to model a large number of differentially expressed genes, is less susceptible to multicollinearity problem, and generalizes better. Deep learning based on transcriptome data has only recently been used to determine the primary effects of gene features that are prognostic to survival of glioblastoma (GBM) or other cancer types [8,18]. However, few studies work on what was learned in these deep learning models. 

Using differentially expressed genes from the GBM-specific TCGA database as inputs, we tested the hypothesis that deep learning can model the relationship between specific genes and the corresponding protein effect to predict patient survival prognosis. Like most of the deep learning models, our model learned a set of features at the last hidden layer, which in this case linearly modulates the survival risk of patients. We hypothesize these features contains the key factors that determine patients’ overall survival for those who have gone through standard of care therapy, from surgery to chemotherapy, without undergoing targeted therapy. 

## 2. Results

### 2.1. Deep Learning Model 

We trained and optimized the deep learning model to generate a network architecture consisting of an input layer feeding to two hidden layers (82 nodes then 27 nodes) and connected to one single output node to predict patient survival prognosis. 

The validation concordance index is 0.69, the corresponding training concordance index is 0.73, and the validation concordance index of each sample partition is evaluated to be with a mean ± 1 SD concordance index of 0.70 ± 0.07. The out-of-sample testing concordance is 0.63 and is within the uncertainty of the validation concordance. The 95% confidence intervals of all testing, validation, and training concordance indexes do not include 0.5.

Important genes that contribute to the overall deep learned model were identified according to the input permutation method. A frequency analysis of genes occurring in the last hidden layer is listed in Table 1. The top 10 ranked genes are either known to be associated with glioblastoma survival, glioblastoma cancer cell migration, or glioblastoma cancer stem cells, or are known to be related to other types of cancer with mechanistic significance. The top 10 genes are *TNR*, *GAD1*, *TMSB15B*, *POSTN*, *SCG3*, *PLA2G2A*, *NNMT*, *CHI3L1*, and *ELAVL4*. The ranked gene importance at each network node is available in Appendix A. The entire frequency analysis table is available in Appendix A. 

In the deep-learned model, no proportional hazard assumption is used. Since the last hidden layer network node outputs are combined linearly with a weight vector to predict patient survival prognosis, the weight vector determines the relative risk from each of the 27 network nodes. Putting these network nodes in a Cox proportional hazard model showed a model concordance of 0.71, with 14 out of 27 network nodes statistically significant using Wald test at *p* < 0.05; only network node 10 is stratified for all network nodes to satisfy the proportional hazard assumption at *p* < 0.05 (Table 2). 

### 2.2. Deep Learning Model Performance Comparison with Penalized Cox Regression Models

Using the training dataset, the k-fold cross validation training resulted in the ridge, adaptive Lasso, and elastic net Cox regression models. These models perform consistently well with concordance indexes at 0.70, 0.69, and 0.70, respectively and are comparable to the deep learning model in performance. In the testing dataset, however, ridge, adaptive Lasso, and elastic net Cox regression models resulted in concordance indexes of 0.58, 0.56, and 0.56, with the 95% confidence intervals of the model concordance index including 0.5. In another word, these models performed close to random and failed to predict survival in the testing dataset, which could be due to the challenge of prediction in a small dataset with 49 patients. This is in sharp contrast to the deep learning model which still performs well in the small testing dataset. 

### 2.3. Network Node Parameters Improved the Baseline Cox Proportional Hazard Model 

A baseline Cox proportional hazard survival model constructed with clinical covariates, including age, gender, Karnofsky Performance Status (KPS), tumor subtypes, and therapy options performed at a concordance value of 0.68. Age, gender, KPS, chemoradiation, and proneural subtype all reached statistical significance using Wald test at *p* < 0.05, with no parameters violating the proportional hazard assumption at *p* < 0.05. Since most patients with the Glioma CpG island methylator phenotype (G-CIMP) mutation also harbor the *IDH1* heterozygous Arg132-to-His (R132H) point mutation, the multicollinearity between these two variables (correlation coefficient = 0.845) led to large standard error on the parameter estimation, affecting statistical significance. 

After adding all deep-learned network nodes, the baseline Cox model (Table 3) concordance index increased from 0.68 to 0.76 with age, KPS, chemoradiation, and proneural subtype statistically significant (Wald test, *p* < 0.05); four risk-increasing network nodes (8, 17, 24, and 25) and four risk-decreasing nodes (4, 10, 16, and 23) were also statistically significant (Wald test, *p* < 0.05), with all covariates satisfying the proportional hazard assumption.

### 2.4. Prognostic Significance Validation of Gene Set with External Data

The chance of a gene deemed important to a small number of network nodes is much higher than it is to a large number of nodes, as shown in Figure 1. We used a probability threshold of *p* < 0.01 and selected a set of genes that are important in at least 10 out of 27 network nodes. These important genes form a 39-gene signature and they are listed in Table 4. Using this gene signature, statistical significance (log-rank test, *p* < 1.5 × 10^−6^) was achieved in seven glioblastoma studies and three low-grade glioma studies in separating the low-risk and high-risk groups with mean ± 1 SD concordance index of 0.79 ± 0.09. The Kaplan–Meier survival curves of the low-risk and high-risk groups in these studies are shown in Figure 2. There are no notable pathway enrichment in the 39-gene signature.

In the discovery process, two particular genes stand out with high hazard ratio (HR) and concordance index (CI) in univariate Cox model, *AQP1* (HR = 3.3, *p* < 0.05, CI = 0.711) and *MEOX2* (HR=2.5, *p* < 0.05, CI = 0.675), both of which are lesser known to be associated with survival in gliomas. Both genes are statistically significant and independently prognostic to survival in the multivariate Cox model (*MEOX2* HR = 2.2, *AQP1* HR = 1.55, both *p* < 0.05, CI = 0.796) in the TCGA GBMLGG dataset.

## 3. Discussion

Prior reports of deep learning model in cancer research [8,18,19] were derived from multiple cancer types to increase sample numbers, but fall short in identifying genes or primary mutations of mechanistic interest. A recent deep feedforward network studies used gene expression feature selection and outcome classification in TCGA breast cancer data as well as in TCGA kidney renal cell carcinoma [20]. In that study, only importance features to the model output were discovered and a number of biologically relevant features were found.

In this paper, we expand the input permutation method for feature importance ranking in deep learning network. Input permutation method is a useful technique for feature importance ranking in machine learning [14] and is broadly applicable to various models. It is usually used to rank feature importance at the model output. Expanding this method to rank features at any hidden layer within a deep learning model opened up many possibilities. It solidifies our understanding of the model and helps in explaining the deep learning model, which is usually considered a black box. In the proposed GBM model, we found that the last hidden layer contains important features that are far more biologically relevant than those obtained from the output layer. 

Compared with gene-signature from previously identified GBM molecular subtypes (classical, proneural, and mesenchymal) [17,21,22], our 39-gene signature has a small overlaps with the 840-gene signatures in classical (8 out of 210), proneural (6 out of 210), and mesenchymal (5 out of 210) subtypes, or 12 genes overlap among all subtypes. The other 27 prognostic genes are likely due to shared biological mechanism(s) among tumor subtypes that are crucial to patient survival. The improvement in model concordance from 0.68 to 0.76 with the addition of deep learned network parameters confirmed the prognostic value of these additional genes in additional to known tumor subtype. It is also quite remarkable that the 39-gene signature learned from one GBM dataset is able to stratify patients in several other low grade glioma datasets as well as other GBM datasets. In addition, two lesser known genes, *AQP1* and *MEOX2*, are discovered to be prognostic to gliomas patients overall survival through the deep learning approach. 

Our deep learning features revealed many genes of interest to glioblastoma stem cells mechanism. For example, both glutamate decarboxylase 1 (*GAD1*) and Chitinase 3 Like 1 (*CHI3L1/YKL-40*) have been recently identified as targets of Notch inhibitors (alpha secretase and gamma secretase inhibitors) in treating glioblastoma stem cells. Notch inhibitors work via Notch binding to YKL-40 and leukemia inhibitory factor (LIF) promotors and increased survival in a GBM stem cell orthotopic mouse model [23]. Epigenetic upregulation of glutamate decarboxylase 1 (*GAD1*) has been shown to program the aggressive features of cancer cell metabolism in brain metastatic microenvironment [24]. Chitinase 3 Like 1 (*CHI3L1/YKL-40*) was also reported previously to be prognostic to glioma patient survival [25] and involved in the angiogenesis, radioresistance, and progression of glioblastoma in vivo [26]. Periostin (*POSTN*) has been shown to impact GBM stem cell tumorigenicity and GBM patient survival [27]. *POSTN* is secreted by glioblastoma stem cells to recruit tumor-associated macrophages in order to promote malignant growth [28] and regulate tumor resistance to anti-angiogenic therapy [29]. Nicotinamide N-methyltransferase (*NNMT*) was also reported to regulate mesenchymal glioblastoma stem cell maintenance by depletion of methionine and shift tumor towards a mesenchymal phenotype and accelerated tumor growth [30]. *NNMT* was reported to be a prognostic marker for glioblastoma [31], inhibiting tumor suppressor protein phosphatase 2 (PP2A) at the epigenome and proteome level and concomitantly activates prosurvival serine/threonine kinases. Receptor tyrosine-protein kinase ErbB-3 (*ERBB3*) is known to mediate glioblastoma cancer stem-like cell resistance to EGFR inhibition [32]. 

Brevican (*BCAN*) which is known to bind to tenascin-R (*TNR*) with high affinity [33], is highly expressed in gliomas, initiating cells’ extracellular niche in human GBM tumors and is expressed by glioma initiating cells in vitro [34]. Though *BCAN* knockdown does not affect glioblastoma initiating cell viability in vitro [34], it promotes glioma cell adhesion and migration in vitro [35] and the knockdown of the gene inhibits both cell motility in vitro and tumorigenicity in vivo [35]. Tenascin-C (*TNC*) and tenascin-R (*TNR*) are two of the three members of the tenascin family of extracellular matrix glycoproteins. TNC (which occurred in four networks) has been shown to promote glioblastoma invasion [36] and is heavily involved in pro-angiogenic and anti-angiogenic signaling in glioblastoma [37], as well as having an impact on survival [38]. A strong *TNR* expression is linked to non-invasive brain tumor (pilocytic astrocytomas) whereas a weak expression is detected in glioblastoma [39]. The exact role of *TNR*, particularly in glioblastoma stem cells extracellular niche, is a subject worth exploring.

On the other hand, ELAV-like RNA binding protein 4 (*ELAVL4*) has been shown to modulate radiation sensitivity in vitro in non-small cell lung cancer [40]. Secretogranin III (*SCG3*) has been shown to be involved in anti-angiogenesis in diabetic retinopathy [41]. Secreted phospholipase A2 group IIA (*PLA2G2A*) was shown to induce phosphorylation of the *EGFR* to induce proliferation through a PKC-dependent pathway in human astrocytoma in vitro [42]. Interestingly, the thymosin beta 15B (*TMSB15B*) is involved in epidermal growth factor-induced migration of prostate cancer cells [43].

Deep learning models that are fully connected and have high dimensional inputs are notoriously difficult to train due to their large number of variables. In our case, the number of variables, about 300,000, is much larger than the number of cases (*n* = 492). Our model is able to generalize on out-of-sample patient cases and has comparable performance in validation concordance index across different data sample splits. Due to the limitation of a single gene chip platform used in this study, its out-of-sample performance on another chip platform may need evaluation.

There are other limitations on the performance of deep learning based on differentially expressed genes. For instance, with 2-fold change cutoff used in this study, we may be artificially removing genes that are prognostic to survival. In addition, the relatively small number of patient samples in this study limits the depth of the model.

Finally, deep learning without explicit biological knowledge or network architecture constraints is not expected to learn biological structure within the data, so care must be taken with biological interpretation. In our case, we identify the prognostic genes by identifying genes with disproportional impact to last hidden layer and the occurrence frequency. Lastly, using deep learning to extract prognostic differential expressed genes for survival prediction provides a flexible way to combine gene expression data with other clinical covariates such as age, KPS, therapy options, and tumor subtypes to enable better patient survival stratification.

## 4. Materials and Methods

### 4.1. Gene Expression Data Analysis

The Cancer Genome Atlas (TCGA) is a publicly repository with patient-derived clinical, imaging, and genomic data that has been deidentified and contains no linkage to patient identifiers, no institutional review board or Health Insurance Portability and Accountability Act approval was required for our study. Microarray data from untreated glioblastoma patients (*n* = 492) were retrieved from TCGA. Gene expression level 1 data from the Affymetrix Human Genome U133A platform were used. The data were processed by software script using R (version 3.2, https://www.R-project.org/, Vienna, Austria), affy [44] and affycoretools [45] packages, and quality control was conducted using affyQCReport package. The probe level data were normalized by gcrma [46] package to control for batch variations, and the probe level expressions are compared to the normal brain tissue group (*n* = 10). Statistically significant gene probe changes were selected with a threshold of *p* < 0.01 with at least 2-fold biological change adjusted for multiple comparisons using the beta uniform mixture model [47]. Genes with a Pearson correlation coefficient higher than 0.8 were represented by one gene to reduce the strong intrinsic correlation. The number of gene probes was reduced to 3581 and used as model inputs.

Clinical data such as age, gender, *MGMT* methylation, G-CIMP, *IDH1/2* mutation, cancer subtype, and therapy information were retrieved from a TCGA publication [22].

### 4.2. Deep Learning Model

The deep learning model was built using Tensorflow 1.3 and Python 3.6 platforms using the deep learning survival modeling framework [48]. Gene expression dataset is randomly partitioned into ten equal partitions, with the first as the testing set, the second as the validation set, and the remaining as training sets. Stratified sampling was used to preserve the survival time distribution among each data partitions to fully capture the heterogeneity from multiple sites. The testing set contained samples from 11 sites whereas the validation set contained samples from 12 sites. The network structure consists of an input layer, one/two hidden layers with rectifier linear unit (Relu) functions, and an output layer with a single node corresponding to the survival prognosis of each patient. The partial likelihood function was used as a loss function and an *L*_2_ penalty was applied on all network weights to prevent overfitting, [49] retaining the advantage of interpretation like Cox’s proportional hazard. Batch-normalization [50] was used in each layer to improve learning stability.

Network structures, including number of hidden layers and number of hidden nodes, are varied to arrive at different models with a maximum of two hidden layers. Hyperparameter tuning on hidden layer(s), nodes and learning rate used concordance index as performance criteria in both the training and validation datasets. A maximum of 1000 epochs were allowed for computation convergence. The optimum hidden layers and nodes were determined by the maximum concordance index in the validation dataset; parameters were stored as a model for testing.

The gene expression dataset was randomly split to training, validation, and testing datasets with ratios of 80%, 10%, and 10%, respectively, while preserving the distribution of survival time to maximize the available datasets for training. The performance variability of the deep learning model was tested by rotating each sample partition as a validation dataset while using the rest for training. After the validation statistics were evaluated, its performance was tested on an out-of-sample data set that was never used in the modeling process.

### 4.3. Deep Learning Model Performance Comparison with Penalized Cox Regression Models

A comparison was made between deep learning model and penalized Cox regression models, including ridge, adaptive Lasso [51], and elastic net [52] using the glmnet package [53] (version 2.0.12).

Cox regression method assumes a semi-parametric hazard form of:$$h_{i}\left(t\right)=h_{0}\left(t\right)e^{x_{i}^{T}\beta }$$
where $h_{i}\left(t\right)$ is the hazard for patient I at time t, $h_{0}\left(t\right)$ is the baseline hazard, and β is a fixed length vector of length p. In penalized Cox regression, models are fitted by maximizing the penalized partial log-likelihood function. The penalized partial log-likelihood function is given by:$$\overset{m}{\underset{i=1}{\prod }}\frac{e^{x_{i}^{T}\beta }}{\sum_{j\in R_{i}}e^{x_{i}^{T}\beta }}-\overset{p}{\underset{j=1}{\sum }}p_{\alpha ,\lambda }\left(\left|\beta_{j}\right|\right)$$
where $p_{\alpha ,\lambda }\left(\left|\cdot \right|\right)$ is the penalty function with tuning parameters *λ* and *α*.

For ridge regression, the penalty function takes the form:$$p_{\alpha ,\lambda }\left(\left|\beta_{j}\right|\right)=\lambda \beta_{j}^{2}$$

For adaptive Lasso regression, the penalty function takes the form:$$p_{\alpha ,\lambda }\left(\left|\beta_{j}\right|\right)=\lambda w_{j}\left|\beta_{j}\right|$$
where $w_{j}=1/\beta_{j0}$. $\beta_{j0}$ the initial estimated of $\beta_{j}$, which in this case is estimated by ridge regression.

For elastic net, the penalty function takes the form:$$p_{\alpha ,\lambda }\left(\left|\beta_{j}\right|\right)=\lambda \left(\alpha \left|\beta_{j}\right|+\left(1-\alpha \right)\frac{1}{2}\beta_{j}^{2}\right)$$
where α∈(0,1].

Model performances were evaluated using concordance index and the same survival time stratified training/validation/testing datasets are used as in the deep learning model for a fair comparison. A 9-fold cross validation was used. A minimum lambda model was chosen as the lamda.1se models are not numerically stable. Feature importance of Lasso, adaptive Lasso, and elastic net methods was identified by the absolute amplitude of regression coefficients.

### 4.4. Impact of Deep Learning Network Features on Baseline Survival Model

The hidden nodes’ outputs are Relu functions that are positive or zero, functioning like an on/off switch that allows effects from a previous level of interacting genes to pass through. Whether these network signatures provide distinct or complementary prognostic value to the baseline survival model was evaluated using Cox proportional hazard model.

A baseline Cox model was constructed from clinical covariates, tumor subtypes, therapy options and genetic mutation/methylation status known to be associated with patient survival. Clinical and genetic mutation/methylation covariates include age, Karnofsky performance scale, gender, *IDH1/2* mutation status, MGMT, G-CIMP methylation status, tumor subtypes (proneural, mesenchymal and classical) as well as chemotherapy/radiation/chemoradiation therapy options. To evaluate the additional prognostic value of deep learning network nodes, they were added to the baseline Cox proportional hazard model. The contribution of these deep learning features in improving survival prediction over the baseline model was evaluated using the concordance index.

### 4.5. Identifying Important Genes in Deep Learning Model

To identify the genes important to survival, we permuted the input genes one gene at a time to break the correlation between the input gene and the output risk [54]. An important gene that contributes significantly to the overall model when permuted across the patient group will impose a significant change to the predicted patient survival, whereas an unimportant gene will not. The process was repeated five times and the average change in patient risk was used. The high impact genes were identified as those that affect predicted patient survival outside the 95% confidence interval of the average change due to single gene permutation.

### 4.6. Prognostic Significance Validation of Gene Set with External Data

To compare the prognostic significance of our gene set in predicting survival in glioblastoma patients, we evaluated it in seven glioblastoma and three low-grade glioma studies (Table 5) using Cox proportional analysis with SurvExpress platform [55]. The samples were split by the median of the prognostic index to designate low-risk and high-risk groups. The top ranked 39 genes, which correspond to *p* < 0.01 or equivalently any gene occurring at least 10 times in the 27 network nodes, were chosen as gene biomarkers. The gene biomarkers were evaluated for survival difference between the low-risk and high-risk groups using log-rank test. The prognostic index is the linear component of the exponential function in the Cox model.

## 5. Conclusions

In conclusion, we discovered that deep learning survival prediction model learned genes that are strongly related to glioblastoma stem cells and/or treatment resistant genes which may be useful to inform patient therapy. Compared with traditional Cox proportional hazard survival models, deep learning networks provide non-redundant prognostic covariates to patient survival even in the presence of strong clinical predictors. Using this approach, we identified many specific genes that are potential biomarkers or therapeutic targets.

## Figures and Tables

**Figure 1 cancers-11-00053-f001:**
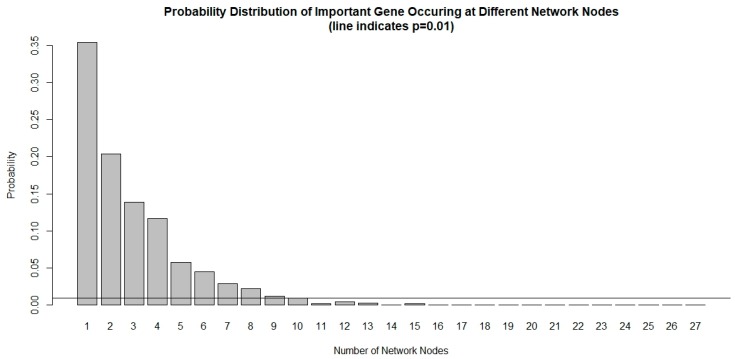
Probability distribution of important genes occurring at different network nodes. It is very rare for an important gene to occur in many nodes that are prognostic to survival. Using a threshold of *p* < 0.01, only genes that occurred at least 10 out of 27 network nodes meet the criteria and are included in the 39-gene signature.

**Figure 2 cancers-11-00053-f002:**
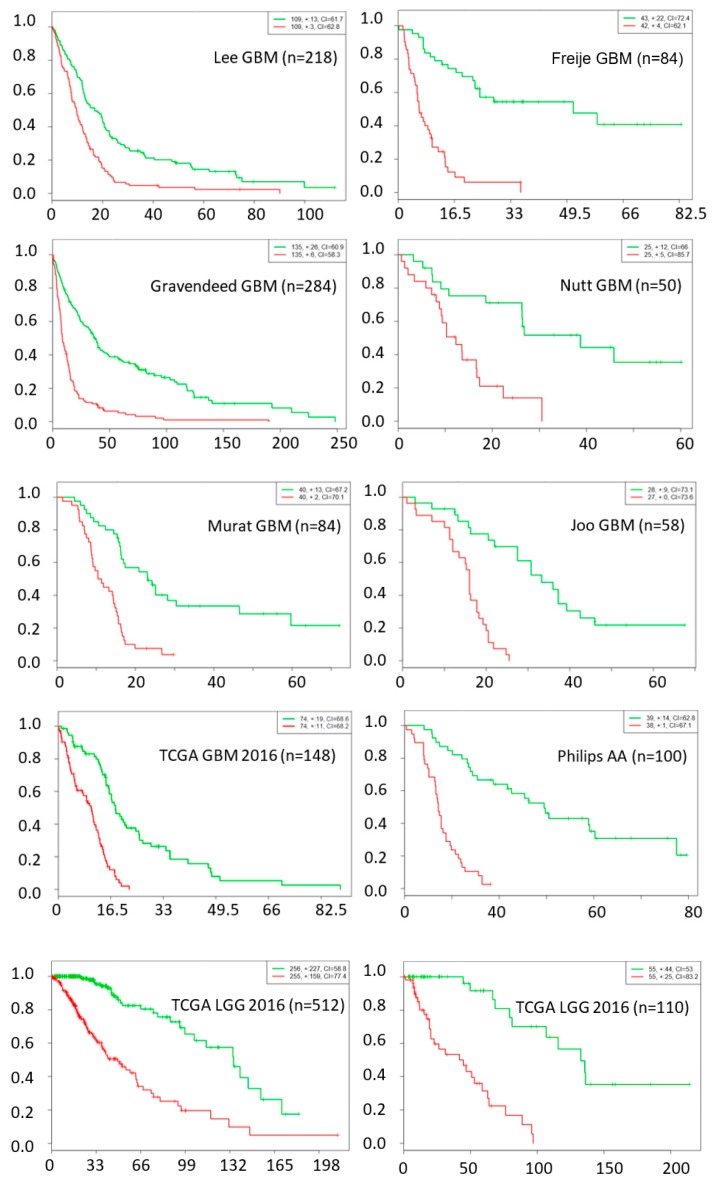
Kaplan–Meier survival fraction versus survival time (months) of the low-risk (green color) and high-risk (red color) groups are well separated using top 39 genes across nine different datasets, including data from seven glioblastoma and three low-grade glioma studies.

**Table 1 cancers-11-00053-t001:** Frequency analysis of important genes in the 27 deep-learned network nodes at the top hidden layer. Only the top 100 frequently occurring genes are listed for brevity.

Gene	Frequency	Gene	Frequency	Gene	Frequency	Gene	Frequency
*TNR*	17	*DPYSL4*	10	*MEG3*	9	*GRB10*	8
*GAD1*	16	*EGFR*	10	*NES*	9	*KDELR3*	8
*TMSB15B*	15	*F13A1*	10	*NPTX2*	9	*KIF1A*	8
*POSTN*	15	*FBN2*	10	*NRXN1*	9	*LSAMP*	8
*SCG3*	15	*NEFM*	10	*NTSR2*	9	*LYPD1*	8
*PLA2G2A*	14	*PTGDS*	10	*PEG3*	9	*MMP9*	8
*NNMT*	13	*RAB6B*	10	*PROM1*	9	*MYT1L*	8
*CHI3L1*	13	*RAPGEF4*	10	*SH3GL3*	9	*NMNAT2*	8
*ELAVL4*	13	*RUNDC3A*	10	*SOX11*	9	*NNAT*	8
*TF*	13	*SERPINA3*	10	*SPOCK1*	9	*NOL4*	8
*UGT8*	13	*SH3GL2*	10	*TMEM35*	9	*NSG1*	8
*AQP1*	12	*SNAP25*	10	*C4B*	8	*PLBD1*	8
*COL6A3*	12	*TCEAL2*	10	*SLC16A3*	8	*RGS1*	8
*ERBB3*	12	*TIMP4*	10	*SOD2*	8	*RGS17*	8
*KCNQ2*	12	*LOC101060835*	9	*AIM1*	8	*RGS4*	8
*LTF*	12	*ADAM22*	9	*ANXA1*	8	*RTN1*	8
*MEOX2*	12	*BCAN*	9	*APOD*	8	*S100A2*	8
*PCDH9*	12	*C1orf61*	9	*ATP2B2*	8	*SLC17A7*	8
*STMN2*	12	*DDX25*	9	*ATP6V1G2*	8	*SRD5A1*	8
*FCGR2B*	11	*ETNPPL*	9	*CFI*	8	*STC1*	8
*FGFR3*	11	*FAM107A*	9	*DSP*	8	*STEAP3*	8
*SLC1A2*	11	*GABRB1*	9	*ENPP2*	8	*STK32B*	8
*CA10*	10	*GDF15*	9	*FCGBP*	8	*TAC1*	8
*CXCL14*	10	*GNAO1*	9	*FUT9*	8	*VSNL1*	8
*CXorf57*	10	*LGI1*	9	*FZD6*	8	*WIF1*	8

**Table 2 cancers-11-00053-t002:** The prognostic value of network node outputs at the top hidden layer are evaluated using Cox proportional hazard model. The hazard ratios (HR) and 95% confidence intervals (95% CI) are listed with corresponding *p*-value. Thirteen network nodes are statistically significant in overall survival prognosis.

Cox Model with Deep Learning Features	HR (95% CI)	*p*-Value
Network Node 0	1.26 (0.98–1.62)	0.0718
Network Node 1	1.13 (0.94–1.36)	0.1996
Network Node 2	1.03 (0.81–1.32)	0.7931
Network Node 3	1.15 (0.94–1.41)	0.1681
Network Node 4	0.73 (0.59–0.89)	0.0022
Network Node 5	0.95 (0.77–1.16)	0.5935
Network Node 6	1.13 (0.88–1.44)	0.341
Network Node 7	1.19 (0.97–1.46)	0.0929
Network Node 8	1.71 (1.40–2.08)	<0.0001
Network Node 9	1.02 (0.81–1.29)	0.8505
Network Node 10		
≥1.6	0.45 (0.25–0.81)	0.0076
<1.6	1	
Network Node 11	0.80 (0.66–0.96)	0.0197
Network Node 12	1.36 (1.11–1.68)	0.0034
Network Node 13	0.93 (0.77–1.14)	0.4994
Network Node 14	1.12 (0.88–1.42)	0.3495
Network Node 15	0.86 (0.67–1.10)	0.2324
Network Node 16	0.57 (0.45–0.72)	<0.0001
Network Node 17	1.35 (1.09–1.67)	0.0056
Network Node 18	0.80 (0.64–1.00)	0.0478
Network Node 19	0.78 (0.64–0.95)	0.0132
Network Node 20	0.91 (0.73–1.15)	0.4437
Network Node 21	1.34 (1.10–1.62)	0.0035
Network Node 22	1.16 (0.95–1.42)	0.1575
Network Node 23	0.77 (0.62–0.97)	0.0281
Network Node 24	1.87 (1.54–2.27)	<0.0001
Network Node 25	1.41 (1.10–1.80)	0.0063
Network Node 26	0.80 (0.65–1.00)	0.0476
Overall Model		<0.0001

**Table 3 cancers-11-00053-t003:** Combined multivariate Cox proportional hazard model including clinical covariates and deep learning network node covariates to predict overall survival (*p*-value < 0.0001). The hazard ratios (HR) and 95% confidence intervals (95% CI) are listed with corresponding *p*-value.

Cox Model with Clinical Covariates and Deep Learning Features	HR (95% CI)	*p*-Value
Age		
≥54 years old	1.50 (1.10–2.03)	0.0098
<54 years old	1	
Gender		
Male	1.25 (0.92–1.68)	0.1542
Female	1	
KPS		
≥60	0.35 (0.17–0.72)	0.0042
<60	1	
Therapy		
Chemoradiation	0.27 (0.12–0.62)	0.0018
Chemotherapy	1.06 (0.35–3.17)	0.9193
Radiation	0.51 (0.22–1.17)	0.1122
Subtype		
Proneural	1.70 (1.01–2.87)	0.0464
Classical	1.26 (0.79–2.01)	0.3311
Mesenchymal	1.41 (0.89–2.25)	0.1462
MGMT Methylated	1.18 (0.85–1.62)	0.3181
G-CIMP Methylated	1.03 (0.35–3.06)	0.9553
R132C/R132G/R132H Mutation	1.08 (0.35–3.31)	0.8986
Network Node 0	1.09 (0.80–1.48)	0.5888
Network Node 1	1.10 (0.87–1.39)	0.4348
Network Node 2	1.15 (0.86–1.55)	0.3407
Network Node 3	1.11 (0.86–1.44)	0.4264
Network Node 4	0.77 (0.60–0.99)	0.0387
Network Node 5	0.85 (0.64–1.12)	0.2558
Network Node 6	1.12 (0.80–1.55)	0.514
Network Node 7	1.12 (0.86–1.44)	0.4041
Network Node 8	1.73 (1.36–2.21)	<0.0001
Network Node 9	1.07 (0.81–1.42)	0.6384
Network Node 10		
≥1.6	0.44 (0.20–0.95)	0.0363
Network Node 11	0.86 (0.67–1.10)	0.2336
Network Node 12	1.27 (0.99–1.65)	0.0645
Network Node 13	1.04 (0.82–1.32)	0.7484
Network Node 14	1.10 (0.82–1.48)	0.5313
Network Node 15	0.79 (0.57–1.11)	0.1711
Network Node 16	0.64 (0.48–0.86)	0.0029
Network Node 17	1.55 (1.16–2.07)	0.0027
Network Node 18	0.79 (0.60–1.05)	0.1049
Network Node 19	0.86 (0.68–1.08)	0.2001
Network Node 20	0.74 (0.54–1.00)	0.0528
Network Node 21		
≥1.6	0.96 (0.55–1.69)	0.8937
Network Node 22	1.22 (0.94–1.58)	0.1327
Network Node 23	0.75 (0.57–1.00)	0.048
Network Node 24	1.66 (1.30–2.12)	<0.0001
Network Node 25	1.49 (1.10–2.01)	0.0101
Network Node 26	0.83 (0.64–1.07)	0.1555
Overall Model		<0.0001

**Table 4 cancers-11-00053-t004:** Gene list of the 39-gene signature selected based on *p* < 0.01 occurrence at the deep-learned network nodes at the top hidden layer.

39-Gene Signature			
*TNR*	*UGT8*	*FGFR3*	*PTGDS*
*GAD1*	*AQP1*	*SLC1A2*	*RAB6B*
*TMSB15B*	*COL6A3*	*CA10*	*RAPGEF4*
*POSTN*	*ERBB3*	*CXCL14*	*RUNDC3A*
*SCG3*	*KCNQ2*	*CXorf57*	*SERPINA3*
*PLA2G2A*	*LTF*	*DPYSL4*	*SH3GL2*
*NNMT*	*MEOX2*	*EGFR*	*SNAP25*
*CHI3L1*	*PCDH9*	*F13A1*	*TCEAL2*
*ELAVL4*	*STMN2*	*FBN2*	*TIMP4*
*TF*	*FCGR2B*	*NEFM*	

**Table 5 cancers-11-00053-t005:** List of glioblastoma studies used in survival prognosis validation using the gene set discovered by deep learning.

Study Datasets	Samples	Source
Lee Nelson Glioblastoma GSE13041 GPL96	218	Lee [56]
Freije Nelson Glioblastoma GSE4412 GPL96	85	Freije [57]
Gravendeed French Glioblastoma GSE16011	284	Gravendeel [58]
Nutt Louis Glioblastoma BROAD	50	Nutt [59]
Murat Hegi Glioblastoma GSE7696	84	Murat [60]
Joo Kim Jin Kim Seol Nam Glioblastoma GSE42669	58	Joo [61]
Philips Aldape Astrocytoma GSE4271 GPL96	100	Phillips [62]
Brain Low Grade Glioma TCGA 2016	110	TCGA
GBM-TCGA June 2016	148	TCGA
LGG-TCGA-Low Grade Gliomas June 2016	512	TCGA

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
