# Peer review of "Prognostic Gene Discovery in Glioblastoma Patients using Deep Learning"

_cancers, 2019, doi:10.3390/cancers11010053_

Round 1

Reviewer 1 Report

Introduction

1.    The first paragraph does nothing to motivate this manuscript

2.    Paragraph 2, there have now been several papers using deep learning for gene expression, which are worth citing.

3.    Well stated hypothesis

Results

4.    No Issue with deep learning model report

5.    Figure 1: Figure texts are difficult to read

Discussion

6.    Overall, well written. I would have liked to see a reference to other deep learning works doing similar analysis

Methods:

7.    The TCGA data set is very heterogeneous and arising from multiple sites. Was there an attempt to ensure the cross-validation was representative of this?

8.    Overall, I thought the methods were well described and thought out

Author Response

Response to Reviewer 1 comments

The authors would like to sincerely thank the anonymous reviewer for their valuable comments and provide the opportunity to improve the quality of the manuscript.  We addressed the concerns in the text and a point-by-point response are listed below for your reference.

Introduction

Point 1.    The first paragraph does nothing to motivate this manuscript

Response 1: The first paragraph is significantly shortened but key references are retained for readers.

Point 2.    Paragraph 2, there have now been several papers using deep learning for gene expression, which are worth citing.

Response 2:  We have added recent deep learning studies using cancer gene expression data in the Introduction section.

Point 3.    Well stated hypothesis

Response 3: Thank you for the comment.

Results

Point 4.    No Issue with deep learning model report

Response 4: Thank you for the comment.

Point 5.    Figure 1: Figure texts are difficult to read

Response 5: The low resolution Figure 1 is replaced with a high resolution version.

Discussion

Point 6.    Overall, well written. I would have liked to see a reference to other deep learning works doing similar analysis

Response 6: The discussion is enriched with a machine learning reference and a recent deep learning studies doing similar analysis.

Methods

Point 7.    The TCGA data set is very heterogeneous and arising from multiple sites. Was there an attempt to ensure the cross-validation was representative of this?

Response 7: This is a great point. We take great care in cross-validation stratified sampling to retain the survival time distribution while capturing the heterogeneity from multiple sites. For example, the testing set contains samples from 11 sites whereas the validation set contains samples from 12 sites. This is clarified in Section 5.2

Point 8.    Overall, I thought the methods were well described and thought out

Response 8: Thank you for the encouragement and useful critiques.

Reviewer 2 Report

This study uses deep learning algorithm to find novel prognostic stratification based on gene expression in glioblastoma and low -grade glioma.

In contrast to previously published studies using similar methods to subtype glial tumors, the authors focus on identifying patterns of gene expression associated with survival. Deep learning applications in patient survival prediction can be of high interest to the broad audience. Being able to distill down to a gene signature that could be routinely tested on RNAseq data from patients’ tumors would be of high potential impact. However, the manuscript does not provide comparisons with previously derived classifiers. The major unanswered question is whether the signatures discovered using the deep learning approach reflect previously identified subtypes (Verhaak et al.), which are associated with survival. In other words: did the use of deep learning uncover unexpected results or did it confirm what was expected? The authors use the TCGA dataset, which contains non-uniformly treated patients. Is the split in survival associated in any way with therapy? This needs to be tested, especially that some patients in this dataset received chemo-radiation, which is known to be superior to radiation or chemotherapy alone.

Since “Cancers” journal is not strictly bioinformatics-oriented, the authors should expand on their rationale and approach. Thus, major revision of the text is recommended.

Additional comments:

1.    Section 2.2: an explanation for different Cox regression models is needed. 

2.    Section 2.4: A table with top 39 genes should be shown. Also, it would help to graphically illustrate the selections process for these genes (ex. Venn diagram).

3.    Section 2.4: The authors discuss selected genes form the 39-gene signature. GSEA or pathway analysis would help see if there are any trends. Were there any genes not previously associated with glioma progression? It would be of interest to know whether deep learning could uncover previously unknown biomarkers.

4.    Is the observed patient stratification independent of therapy that the patients received?

5.    The discussion section in the current version focuses on function of genes selected form the signature. It would be of interest to discuss the previously published approaches to deep learning-based stratification attempts.

6.    Line 178: “Lastly, using differential expressed genes for survival  prediction is limited as it is generally accepted that patients’ _genes do not wholly determine survival.” This statement should be edited. If the authors intention was to point that genetic determinants and gene expression do not always match, it should be stated. Or if the authors wanted to discuss other survival predictors, such as age or KPS, it should be included.

Author Response

Response to Reviewer 2 comments

The authors would like to sincerely thank the anonymous reviewer for their valuable comments and provide the opportunity to improve the quality of the manuscript.  We addressed the concerns in the text and a point-by-point response are listed below for your reference.

This study uses deep learning algorithm to find novel prognostic stratification based on gene expression in glioblastoma and low -grade glioma.

In contrast to previously published studies using similar methods to subtype glial tumors, the authors focus on identifying patterns of gene expression associated with survival. Deep learning applications in patient survival prediction can be of high interest to the broad audience. Being able to distill down to a gene signature that could be routinely tested on RNAseq data from patients’ tumors would be of high potential impact.

Point 1.    However, the manuscript does not provide comparisons with previously derived classifiers. The major unanswered question is whether the signatures discovered using the deep learning approach reflect previously identified subtypes (Verhaak et al.), which are associated with survival. In other words: did the use of deep learning uncover unexpected results or did it confirm what was expected?

Response 1: Deep learning uncovers many genes that are different from previous identified subtypes. This is clarified in the third paragraph in the Discussion section. In addition, most GBM subtypes associated with survival did not reach statistical significance (p<0.05) when clinical covariates and deep learned gene features are used in the multivariate Cox regression model.

Point 2.    The authors use the TCGA dataset, which contains non-uniformly treated patients. Is the split in survival associated in any way with therapy? This needs to be tested, especially that some patients in this dataset received chemo-radiation, which is known to be superior to radiation or chemotherapy alone.

Response 2: Thank you for the great suggestions. The type of therapy and cancer subtypes are now included in the model in Table 3.  Indeed, chemoradiation and proneural subtype are significant factors and is reflected in the Section 2.3.

Point 3.    Since “Cancers” journal is not strictly bioinformatics-oriented, the authors should expand on their rationale and approach. Thus, major revision of the text is recommended.

Response 3: Section 1 and Section 5.3 have been revised to include more details.

Additional comments:

Point 4.    Section 2.2: an explanation for different Cox regression models is needed.

Response 4: We have expanded Section 5.3 to include detailed explanation of different Cox regression models. Section 2.2 is also clarified the performance of different Cox regression models.

Point 5.    Section 2.4: A table with top 39 genes should be shown. Also, it would help to graphically illustrate the selections process for these genes (ex. Venn diagram).

Response 5: The selection process for these genes are explained graphically in a new Figure 1 using probability threshold. The selection process are explained in more details in Section 2.4. A new Table 4 now shows the top 39 genes.

Point 6.    Section 2.4: The authors discuss selected genes form the 39-gene signature. GSEA or pathway analysis would help see if there are any trends. Were there any genes not previously associated with glioma progression? It would be of interest to know whether deep learning could uncover previously unknown biomarkers.

Response 6: Although there is no notable pathway enrichment of the 39-gene signature by IPA/DAVID, two particular genes stand out with high concordance index (CI) in univariate Cox model, AQP1 (HR=3.3, p<0.001, CI=0.711) and MEOX2 (HR=2.5, p<0.001, CI=0.675), both of which are lesser known to be associated with survival in gliomas. Both genes are statistically significant and independently prognostic to survival in multivariate Cox model (MEOX2 HR=2.2, AQP1 HR=1.55, both p<0.001, CI=0.796) in the TCGA GBMLGG dataset.  This is updated in Section 2.4. 

Point 7.    Is the observed patient stratification independent of therapy that the patients received?

Response 7: In the baseline Cox model, proneural subtype and chemoradiation are significant factors in patient stratification. When deep learning network parameters are added to the Cox model, model performance improved significantly. Both proneural subtype and chemoradiation remain statistically significant along with age, KPS, and eight different network parameters. This is reflected in Section 2.3 and Table 3.

Point 8.    The discussion section in the current version focuses on function of genes selected form the signature. It would be of interest to discuss the previously published approaches to deep learning-based stratification attempts.

Response 8: The discussion is enriched with a machine learning reference and a recent deep learning studies doing similar analysis by looking at the outside of the model output. We highlighted our approach by looking inside the deep learning model to find out what was learned by the model.

Point 9.    Line 178: “Lastly, using differential expressed genes for survival prediction is limited as it is generally accepted that patients’ genes do not wholly determine survival.” This statement should be edited. If the authors’ intention was to point that genetic determinants and gene expression do not always match, it should be stated. Or if the authors wanted to discuss other survival predictors, such as age or KPS, it should be included.

Response 9: The statement is clarified to highlight the flexibility of deep learning gene feature that can be easily combined with other clinical covariates for survival stratification.

Round 2

Reviewer 2 Report

The authors have answered all questions sufficiently.